# CRYPTEN: Secure Multi-Party Computation Meets Machine Learning

**Brian Knott**    **Shobha Venkataraman**    **Awni Hannun**
**Shubho Sengupta**    **Mark Ibrahim**    **Laurens van der Maaten**
Facebook AI Research
{brianknott,shobha,awni,ssengupta,marksibrahim,lvdmaaten}@fb.com

## Abstract

Secure multi-party computation (MPC) allows parties to perform computations on data while keeping that data private. This capability has great potential for machine-learning applications: it facilitates training of machine-learning models on private data sets owned by different parties, evaluation of one party's private model using another party's private data, *etc.* Although a range of studies implement machine-learning models via secure MPC, such implementations are not yet mainstream. Adoption of secure MPC is hampered by the absence of flexible software frameworks that "speak the language" of machine-learning researchers and engineers. To foster adoption of secure MPC in machine learning, we present CRYPTEN: a software framework that exposes popular secure MPC primitives via abstractions that are common in modern machine-learning frameworks, such as tensor computations, automatic differentiation, and modular neural networks. This paper describes the design of CRYPTEN and measure its performance on state-of-the-art models for text classification, speech recognition, and image classification. Our benchmarks show that CRYPTEN's GPU support and high-performance communication between (an arbitrary number of) parties allows it to perform efficient private evaluation of modern machine-learning models under a *semi-honest* threat model. For example, two parties using CRYPTEN can securely predict phonemes in speech recordings using Wav2Letter [17] faster than real-time. We hope that CRYPTEN will spur adoption of secure MPC in the machine-learning community.

## 1   Introduction

Secure multi-party computation (MPC; [30, 69]) allows parties to collaboratively perform computations on their combined data sets without revealing the data they possess to each other. This capability of secure MPC has the potential to unlock a variety of machine-learning applications that are currently infeasible because of data privacy concerns. For example, secure MPC can allow medical research institutions to jointly train better diagnostic models without having to share their sensitive patient data [27] or allow social scientists to analyze gender wage gap statistics without companies having to share sensitive salary data [42]. The prospect of such applications of machine learning with rigorous privacy and security guarantees has spurred a number of studies on machine learning via secure MPC [38, 41, 48, 58, 63, 66, 67]. However, at present, adoption of secure MPC in machine learning is still relatively limited considering its wide-ranging potential. One of the main obstacles to widespread adoption is that the complexity of secure MPC techniques puts them out of reach for most machine-learning researchers, who frequently lack in-depth knowledge of cryptographic techniques.

To foster the adoption of secure MPC techniques in machine learning, we present CRYPTEN: a flexible software framework that aims to make modern secure MPC techniques accessible to machine-learning researchers and developers without a background in cryptography. Specifically, CRYPTEN

35th Conference on Neural Information Processing Systems (NeurIPS 2021).

provides a comprehensive tensor-computation library in which all computations are performed via secure MPC. CRYPTEN's API closely follows the API of the popular PyTorch framework for machine learning [54, 55], which makes it easy to use for machine-learning practitioners. For example, it provides automatic differentiation and a modular neural-network package. CRYPTEN assumes an *semi-honest* threat model [30, §2.3.2] and works for an arbitrary number of parties. To make private training and inference efficient, CRYPTEN off-loads computations to the GPU and uses high-performance communication libraries to implement interactions between parties.

The paper presents: (1) an overview of CRYPTEN's design principles; (2) a description of the design of CRYPTEN and of the secure MPC protocols implemented; (3) a collection of benchmark experiments using CRYPTEN to run private versions of state-of-the-art models for text classification, speech recognition, and image classification; and (4) a discussion of open problems and a roadmap for the further development of CRYPTEN. Altogether, the paper demonstrates that CRYPTEN's flexible, PyTorch-like API makes private inference and training of modern machine-learning models easy to implement and efficient. For example, CRYPTEN allows two parties to privately classify an image [26, 35] in 2-3 seconds, or to securely make phoneme predictions for 16kHz speech recordings [17] faster than real-time. We hope that CRYPTEN's promising performance and ease-of-use will foster the adoption of secure MPC by the machine-learning community, and pave the way for a new generation of secure and private machine-learning systems.

## 2  Related Work

CRYPTEN is part of a large body of work that develops secure MPC protocols for machine learning; see Appendix **??**. Most closely related to our work is CryptGPU [63], which implements an 2-out-of-3 replicated secret sharing protocol [4, 37] *on top of* CRYPTEN. Like CRYPTEN, CryptGPU provides security against *semi-honest* corruption, but it is limited to the three-party setting. CryptGPU is one of several protocols optimized for the three-party setting. For example, Falcon [67] implements a *maliciously secure* three-party MPC protocol, combining techniques from SecureNN [66] and ABY3 [48]. Falcon allows evaluation and training of convolutional networks such as AlexNet [40] and VGG [62]. Other systems that work in this setting include Astra [16], Blaze [56], and CrypTFlow [41].

There also exists a family of two-party systems that, like CRYPTEN, assume a semi-honest threat model. These systems include Gazelle [38], Chameleon [58], EzPC [15], MiniONN [45], SecureML [49], PySyft [60], and Delphi [47]. XONN [59] also works in the two-party setting but provides malicious security. Compared to these systems, CRYPTEN provides a more flexible machine-learning focused API[1] that supports reverse-mode automatic differentiation, implements a rich set of functions, and natively runs on GPUs. Moreover, CRYPTEN supports a wider range of use cases by working with an arbitrary number of parties, and make communication between parties efficient via communication primitives that were optimized for high-performance distributed computing.

## 3  Design Principles

In the development of CRYPTEN, we adopted the following two main design principles:

**Machine-learning first API.** CRYPTEN has a general purpose, machine-learning first API design. Most other secure MPC frameworks [34] adopt an API that stays close to the underlying MPC protocols. This hampers adoption of these frameworks in machine learning, for example, because they do not natively support tensor operations (but only scalar operations) and because they lack features that machine-learning researchers have come to expect, such as automatic differentiation. Instead, CRYPTEN implements the tensor-computation API of the popular PyTorch machine-learning framework [54], implements reverse-mode automatic differentiation, provides a modular neural-network package with corresponding learning routines, and supports GPU computations. We aim to allow developers to transition code from PyTorch to CRYPTEN by changing a single Python `import`.

**Eager execution.** CRYPTEN adopts an imperative programming model. This is different from existing MPC frameworks, which generally implement compilers for their own domain-specific languages [34]. While compiler approaches have potential performance benefits, they slow down the

---

[1]CrypTFlow [41] also provides such an API by integrating deeply with TensorFlow [1], but unlike CRYPTEN, it does not support PyTorch's eager execution model [55] or GPU support.

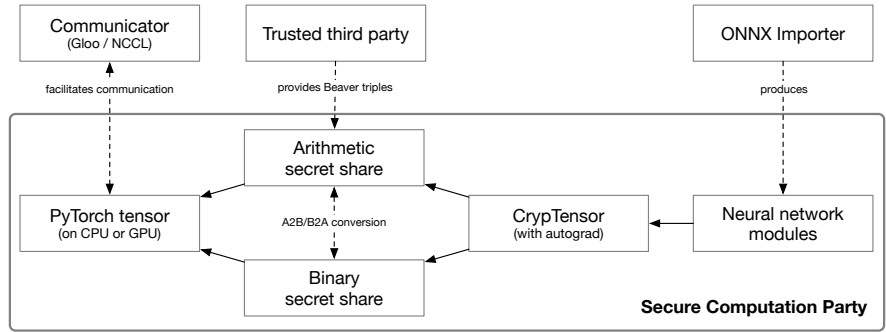

Figure 1: High-level overview of the design of CRYPTEN. See text in Section 4 for details.

development cycle, make debugging harder, and prevent users from using arbitrary host-language constructs [3]. Instead, CRYPTEN follows the recent trend in machine learning away from graph compilers [1] to frameworks that eagerly execute computations [3, 55], providing a better developer experience. Yet, CRYPTEN is performant because it implements state-of-the-art secure MPC protocols (for settings with arbitrary number of parties), because it uses PyTorch's highly optimized tensor library for most computations, because computations can be off-loaded to the GPU, and because it uses communication libraries that were optimized for high-performance distributed computing.

## 4   Design Overview

Figure 1 gives an overview of CRYPTEN's design. Parties perform computations using efficient PyTorch tensor operations. Because secure MPC computations are integer computations that are not natively supported on GPUs, CRYPTEN maps between integer and floating-point computations on GPUs; see Section 5.3. The multi-party computations are implemented on arithmetic and binary secret shares [22, 32]; see Section 5.1. Whereas many computations can be performed directly on arithmetic secret shares, others require conversion between arithmetic and binary secret shares (A2B) and back (B2A); see Section 5.2. Some multi-party computations require interaction between parties via a *communicator* that employs the high-performance communication primitives in Gloo [31] and NCCL [51]. Some multi-party computations require Beaver triples [7], which are supplied by a *trusted third party* (TTP).[2]

```python
import crypten, torch

# set up communication and sync random seeds:
crypten.init()

# secret share tensor:
x = torch.tensor([1.0, 2.0, 3.0])
x_enc = crypten.cryptensor(x, src=0)

# reveal secret shared tensor:
x_dec = x_enc.get_plain_text()
assert torch.all_close(x_dec, x)

# add secret shared tensors:
y = torch.tensor([2.0, 3.0, 4.0])
y_enc = crypten.cryptensor(y, src=0)
xy_enc = x_enc + y_enc
xy_dec = xy_enc.get_plain_text()
assert torch.all_close(xy_dec, x + y)
```

Figure 2: Example of secret-sharing tensors, revealing tensors, and private addition in CRYPTEN.

All secure computations are wrapped in a `CrypTensor` object that implements the PyTorch tensor API and that provides reverse-mode automatic differentiation (autograd) to enable gradient-based training of arbitrary (deep) learning models. Figure 2 illustrates `CrypTensor` creation, *i.e.*, how tensors are secret-shared and revealed, as well as a simple computation (addition). Note that each party involved in the multi-party computation executes the same code. Whenever communication between the parties is required (*e.g.*, as part of private multiplications), the communication acts as a synchronization point between the parties. The `crypten.init()` call is required once to establish the communication channel. In the example, the input tensor for the creation of the arithmetic secret share is provided party `src=0`, which indicates the rank[3] of the party that supplies the data to be secret-shared (the other parties executing this code may provide `None` as input).

---

[2]CRYPTEN adopts a trusted third party for generating Beaver triples for efficiency reasons, but we are planning to add TTP-free solutions based on additive homomorphic encryption [52] or oblivious transfer [39].

[3]CRYPTEN relies on MPI primitives for communication: each party knows their rank and the world size.

To enable deep-learning use cases, CRYPTEN allows implementing neural networks following PyTorch's API. Figure 3 shows how to create and encrypt neural networks and how to use automatic differentiation in CRYPTEN. The example assumes that some training `sample` and the associated `target` label are provided by the party with rank 0 (note the value of `src`). As illustrated by the example, CRYPTEN's API closely follows that of PyTorch. Indeed, it is possible to write a single training loop that can be used to train models using CRYPTEN or PyTorch without code changes. This makes it easy to adapt PyTorch code to use secure MPC for its computations, and it also makes debugging easier. The appendix presents a table listing all tensor functions that `CrypTensor` implements.

To enable interoperability with existing machine-learning platforms, neural networks can be imported into CRYPTEN via ONNX. Figure 4 shows how a PyTorch model is imported into CRYPTEN. The example illustrates how CRYPTEN makes private inference with a ResNet-18 easy. The example in the figure also demonstrates CRYPTEN's GPU support. One caveat is that all parties must use the same type of device (*i.e.*, CPU or GPU) for computations.

## 5 Secure Computations

To facilitate secure computations, CRYPTEN implements arithmetic secret sharing [22, 23] and binary secret sharing [32], as well as conversions between these two types of sharing [24]. Arithmetic secret sharing is particularly well-suited for operations that are common in modern machine-learning models, such as matrix multiplications and convolutions. Binary secret sharing is required for evaluating certain other common functions, such as rectified linear units. We provide a high-level overview of CRYPTEN's secure computation protocol here; a detailed description is presented in the appendix.

```
import crypten.optimizer as optimizer
import crypten.nn as nn

# create model, criterion, and optimizer:
model_enc = nn.Sequential(
    nn.Linear(sample_dim, hidden_dim),
    nn.ReLU(),
    nn.Linear(hidden_dim, num_classes),
).encrypt()
criterion = nn.CrossEntropyLoss()
optimizer = optimizer.SGD(
    model_enc.parameters(), lr=0.1, momentum=0.9,
)

# perform prediction on sample:
target_enc = crypten.cryptensor(target, src=0)
sample_enc = crypten.cryptensor(sample, src=0)
output_enc = model_enc(sample_enc)

# perform backward pass and update parameters:
model_enc.zero_grad()
loss_enc = criterion(output_enc, target_enc)
loss_enc.backward()
optimizer.step()
```

Figure 3: Example using neural networks and automatic differentiation in CRYPTEN.

```
import torchvision.datasets as datasets
import torchvision.models as models
import torchvision.transforms as transforms

# download and set up ImageNet dataset:
transform = transforms.ToTensor()
dataset = datasets.ImageNet(
    imagenet_folder, transform=transform,
)

# secret share pre-trained ResNet-18 on GPU:
model = models.resnet18(pretrained=True)
model_enc = crypten.nn.from_pytorch(
    model, dataset[0],
).encrypt().cuda()

# perform inference on secret-shared images:
for image in dataset:
    image_enc = crypten.cryptensor(image).cuda()
    output_enc = model_enc(image_enc)
    output = output_enc.get_plain_text()
```

Figure 4: Private inference on secret-shared images using a secret-shared ResNet-18 model on GPU.

### 5.1 Secret Sharing

**Arithmetic secret sharing** shares a scalar value $x \in \mathbb{Z}/Q\mathbb{Z}$, where $\mathbb{Z}/Q\mathbb{Z}$ denotes a ring with $Q$ elements, across parties $p \in \mathcal{P}$. We denote the sharing of $x$ by $[x] = \{[x]_p\}_{p\in\mathcal{P}}$, where $[x]_p \in \mathbb{Z}/Q\mathbb{Z}$ indicates party $p$'s share of $x$. The shares are constructed such that their sum reconstructs the original value $x$, that is, $x = \sum_{p\in\mathcal{P}}[x]_p \mod Q$. To share a value $x$, the parties generate a pseudorandom zero-share [18] with $|\mathcal{P}|$ random numbers that sum to 0. The party that possesses the value $x$ adds $x$ to their share and discards $x$. We use a fixed-point encoding to obtain $x$ from a floating-point value, $x_R$. To do so, we multiply $x_R$ with a large scaling factor $B$ and round to the nearest integer: $x = \lfloor Bx_R \rceil$, where $B = 2^L$ for some precision of $L$ bits. To decode a value, $x$, we compute $x_R \approx x/B$.

**Binary secret sharing** is a special case of arithmetic secret sharing that operates within the binary field $\mathbb{Z}/2\mathbb{Z}$. A binary secret share, $\langle x \rangle$, of a value $x$ is formed by arithmetic secret shares of the bits of $x$, setting $Q = 2$. Each party $p \in \mathcal{P}$ holds a share, $\langle x \rangle_p$, such that $x = \bigoplus_{p\in\mathcal{P}}\langle x \rangle_p$ is satisfied.

**Conversion from $[x]$ to $\langle x \rangle$** is implemented by having the parties create a binary secret share of their $[x]_p$ shares, and summing the resulting binary shares. Specifically, the parties create a binary secret share, $\langle [x]_p \rangle$, of all the bits in $[x]_p$. Subsequently, the parties compute $\langle x \rangle = \sum_{p \in \mathcal{P}} \langle [x]_p \rangle$ using a carry-lookahead adder in $\log_2(|\mathcal{P}|) \log_2(L)$ communication rounds [14, 21].

**Conversion from $\langle x \rangle$ to $[x]$** is achieved by computing $[x] = \sum_{b=1}^{B} 2^b \left[ \langle x \rangle^{(b)} \right]$, where $\langle x \rangle^{(b)}$ denotes the $b$-th bit of the binary share $\langle x \rangle$ and $B$ is the total number of bits in the shared secret, $\langle x \rangle$. To create an arithmetic share of a bit, the parties use secret shares, $\left( [r^{(b)}], \langle r^{(b)} \rangle \right)$, of random bits $r^{(b)}$. The random bits are provided by the TTP, but we plan to add an implementation that generates them off-line via oblivious transfer [39]. The parties use $\langle r^{(b)} \rangle$ to mask $\langle x \rangle^{(b)}$ and reveal the resulting masked bit $z^{(b)}$. Subsequently, they compute $\left[ \langle x \rangle^{(b)} \right] = \left[ r^{(b)} \right] + z^{(b)} - 2 \left[ r^{(b)} \right] z^{(b)}$.

## 5.2 Secure Computation

Arithmetic and binary secret shares have homomorphic properties that can be used to implement secure computations. All computations in CRYPTEN are based on private addition and multiplication.

**Private addition** of two arithmetically secret shared values, $[z] = [x] + [y]$, is implemented by having each party $p$ sum their shares of $[x]$ and $[y]$: each party $p \in \mathcal{P}$ computes $[z]_p = [x]_p + [y]_p$.

**Private multiplication** is implemented using random Beaver triples [7], $([a], [b], [c])$ with $c = ab$, that are provided by the TTP. The parties compute $[\epsilon] = [x] - [a]$ and $[\delta] = [y] - [b]$, and decrypt $\epsilon$ and $\delta$ without information leakage due to the masking. They compute the result $[x][y] = [c] + \epsilon[b] + [a]\delta + \epsilon\delta$, using trivial implementations of addition and multiplication of secret shares with public values.

**Linear functions** are trivially implemented as combinations of private addition and multiplication. This allows CRYPTEN to compute dot products, outer products, matrix products, and convolutions.

**Non-linear functions** are implemented using standard approximations that only require private addition and multiplication. Specifically, CRYPTEN evaluates exponentials using a limit approximation, logarithms using Householder iterations [36], and reciprocals using Newton-Rhapson iterations. This allows CRYPTEN to implement functions that are commonly used in machine-learning models, including the sigmoid, softmax, and logistic-loss functions, as well as their gradients.

**Comparators** are implemented using a function that evaluates $[z < 0]$ by: (1) converting $[z]$ to a binary secret-share $\langle z \rangle$; (2) computing its sign bit, $\langle b \rangle = \langle z \rangle >> (L-1)$; and (3) converting the resulting bit to an arithmetic sharing $[b]$. This function allows CRYPTEN to implement arbitrary comparators. For example, it evaluates $[x < y]$ by computing $[z] = [x] - [y]$ and evaluating $[z < 0]$. Similarly, CRYPTEN can evaluate: (1) the sign function via $\text{sign}([x]) = 2[x > 0] - 1$; (2) the absolute value function via $|[x]| = [x]\,\text{sign}([x])$; and (3) rectified linear units via $\text{ReLU}([x]) = [x][x > 0]$. CRYPTEN also supports multiplexing; to do so, it evaluates $[c \,?\, x : y] = [c][x] + (1 - [c])[y]$.

**Lemma 1.** *The* CRYPTEN *secure-computation protocol is secure against information leakage against any static passive adversary corrupting up to $|\mathcal{P}| - 1$ of the $|\mathcal{P}|$ parties involved in the computation.*

The proof of this lemma follows trivially from [9, 11, 21, 24], and is given in the appendix. We adopt a protocol that provides security under a *semi-honest* threat model because it enables a wide range of use cases of secure machine learning, whilst being more efficient than maliciously secure protocols.

## 5.3 Off-loading Computations to the GPU

Hardware acceleration via GPUs is a critical component for training and inference in modern machine-learning models. Akin to frameworks such as PyTorch [55] and TensorFlow [1], CRYPTEN can off-load computations to the GPU. On the GPU, it uses highly-optimized implementations for a range of functions that are provided by CUDA libraries such as cuBLAS [19] and cuDNN [20].

Unfortunately, these libraries are designed for computations on floating-point numbers and do not support the integer types required to perform computations on $L$-bit fixed-point numbers. Akin to [63], we circumvent this problem by observing that for all integers $a, b \in \mathbb{Z} \cap [-2^{26}, 2^{26}]$, we can compute the product $ab$ using 64-bit floating-point representations and still recover the correct value over the integers. Specifically, CRYPTEN splits each 64-bit variable into four components, $a = a_0 + 2^{16}a_1 + 2^{32}a_2 + 2^{48}a_3$, where each $a_i$ represents a 16-bit integer component. We compute a product $ab$ of 64-bit integers by summing 10 pairwise products of their 16-bit components.

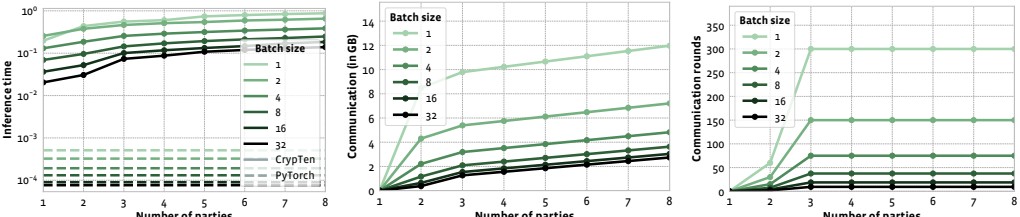

Figure 5: Benchmarks for inference with text-sentiment classification model on GPUs in CRYPTEN and PyTorch. **Left:** Average wall-clock time per sample (in seconds). **Middle:** Number of bytes communicated per sample, per party (in GB). **Right:** Number of communication rounds per sample.

The pairwise products of the 16-bit components are computed in parallel using highly optimized floating-point CUDA kernels. The same approach is used for matrix multiplications and convolutions. CRYPTEN further optimizes this approach by splitting into only 3 components of 22-bits each when possible, which reduces the number of pairwise products required to 6 (see [63, Remark II.1]).

## 6 Benchmarks

To measure the performance of CRYPTEN, we performed experiments on three tasks: (1) text classification using a linear model that learns word embeddings; (2) speech recognition using the Wav2Letter model [17]; and (3) image classification using residual networks [35] and vision transformers [26]. Because of space constraints, we focus on private inference using a secret-shared model on secret-shared data here, but our benchmark results with private training are very similar.

We performed benchmark experiments on a proprietary cluster, testing inference on both CPUs (Intel Skylake 18-core 1.6GHz) and GPUs (nVidia P100). We set the number of OpenMP threads to 1 in all benchmarks. All experiments were performed with the parties running in separate processes on a single machine. For GPU experiments, each party was assigned its own GPU. Although this setup is faster than a scenario in which each party operates its own machine,[4] we believe our benchmark results provide a good sense of CRYPTEN's performance. We average computation times over 30 batches, excluding the computation on the first batch as that computation may include CuDNN benchmarking. Code reproducing the results of our experiments is available on `https://crypten.ai`.

In our benchmarks, we focus on comparing (ciphertext) CRYPTEN computation with (plaintext) PyTorch computation. We refer the reader to [33, 63] for benchmarks that compare CRYPTEN to other secure MPC frameworks. Specifically, [33] finds CRYPTEN is 11-18× faster than PySyft [60] and approximately 3× faster than TF-Trusted [13] in MNIST classification [43] on CPU.

### 6.1 Text Classification

We performed text-sentiment classification experiments on the Yelp review dataset [70] using a model that consists of a linear layer operating on word embeddings. The embedding layer contains 32-dimensional embeddings of $519,820$ words, and the linear layer produces a binary output indicating the sentiment of the review. We evaluated the model on GPUs, varying the batch size and the number of parties participating. The normalized mean squared error ($\|\mathbf{x}-\mathbf{y}\|^2/\|\mathbf{x}\|^2$) between the output of the CRYPTEN model and that of its PyTorch counterpart was smaller than $4 \cdot 10^{-4}$ in all experiments.

Figure 5 presents the results of our experiments. The figure shows inference time *per sample* (in seconds) as a function of the number of parties involved in the computation for varying batch sizes (left); the amount of communication required per sample, *per party* (in GB); and the number of communication rounds required per sample. We include results in which the number of parties is 1: herein, we run the CRYPTEN protocol but involve no other parties, which implies that the single party is running the protocol on unencrypted data. One-party results allow us to bisect different sources of computational overhead: specifically, they separate overhead due to communication from overhead due to fixed-point encoding, function approximations, and (lack of) sparse-matrix operations.

---

[4]Communication between GPUs in two machines connected via InfiniBand has approximately $20\times$ lower throughput than communication between two GPUs in the same machine via NVLink (25GB/s versus 600GB/s).

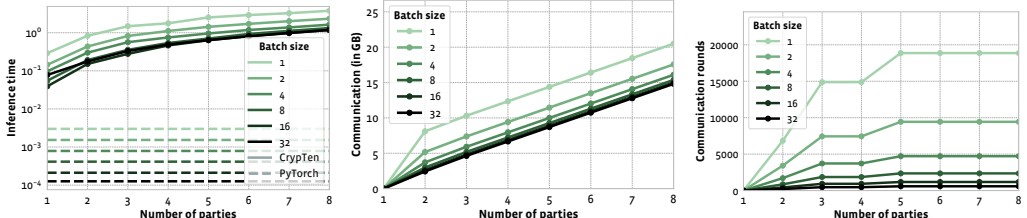

Figure 6: Benchmarks for inference with Wav2Letter model on GPUs in CRYPTEN and PyTorch. **Left:** Average wall-clock time per sample (in seconds). **Middle:** Number of bytes communicated per sample, per party (in GB). **Right:** Number of communication rounds per sample.

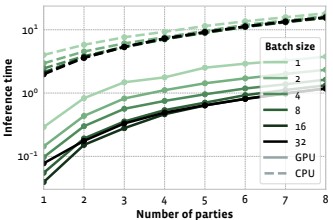 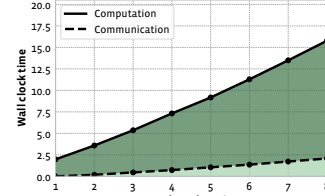 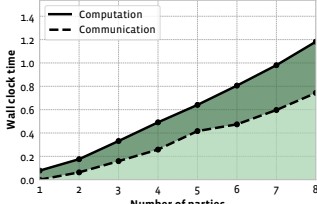

Figure 7: Wall-clock time per sample (in sec.) for Wav2Letter inference on CPUs and GPUs.

Figure 8: Average wall-clock time per sample (in seconds) for communication and computation during inference with Wav2Letter model on CPU (**left**) and GPU (**right**).

The results in Figure 5 show that CRYPTEN is about 2.5–3 orders of magnitude slower than PyTorch in text-sentiment classification, depending on the number of parties involved. Most computational overhead is the word embedding layer: whereas PyTorch can evaluate this layer efficiently via a sparse matrix multiplication, CRYPTEN cannot do sparse lookups as they would reveal information on the encrypted input. Instead, CRYPTEN performs a full matrix multiplication between the word-count vector and the embedding matrix. Yet, text sentiment predictions are quite fast in CRYPTEN: inference takes only $0.03$ seconds per sample in the two-party setting with a batch size of $32$.

The results also show that increasing the batch size is an effective way to reduce inference time and communication per sample. The number of communication rounds is independent of the batch size, which means communication rounds can be amortized by using larger batch sizes. The number of bytes communicated is partly amortized as well because the size of weight tensors (*e.g.*, in linear layers) does not depend on batch size. The results also show that whereas the number of communication rounds increases when moving from two-party to three-party computation, it remains constant afterwards. The larger number of communication rounds for three-party computation stems from the public division protocol, which requires additional communication rounds when more than two parties are involved to prevent wrap-around errors (see the appendix for details).

### 6.2 Speech Recognition

We performed speech-recognition experiments using Wav2Letter [17] on the LibriSpeech dataset [53]. The LibriSpeech dataset contains $16$ kHz audio clips represented as a waveform ($16,000$ samples per second). Because the audio clips vary in length, we clip all of them to $1$ second for the benchmark. Wav2Letter is a network with 13 convolutional layers using rectified linear unit (ReLU; [50]) activations.[5] The network operates directly on the waveform input, predicting one of 29 labels (26 letters plus 3 special characters). The first two layers use a filter size of $250$ (with stride $160$) and $48$ (stride $2$). The next seven layers use filter size $7$, followed by two layers with filter size $32$ and $1$ (all with stride $1$). All layers except the last two have $250$ channels. The last two layers have $2,000$ channels.

The results in Figure 6 show that CRYPTEN is about 2.5–3 orders of magnitude slower than PyTorch depending on the number of parties involved. For Wav2Letter, the overhead is largely due to the ReLU layers in the network: evaluating a ReLU function requires a comparison, which involves a

---

[5]We used the reference implementation of Wav2Letter in `torchaudio`.

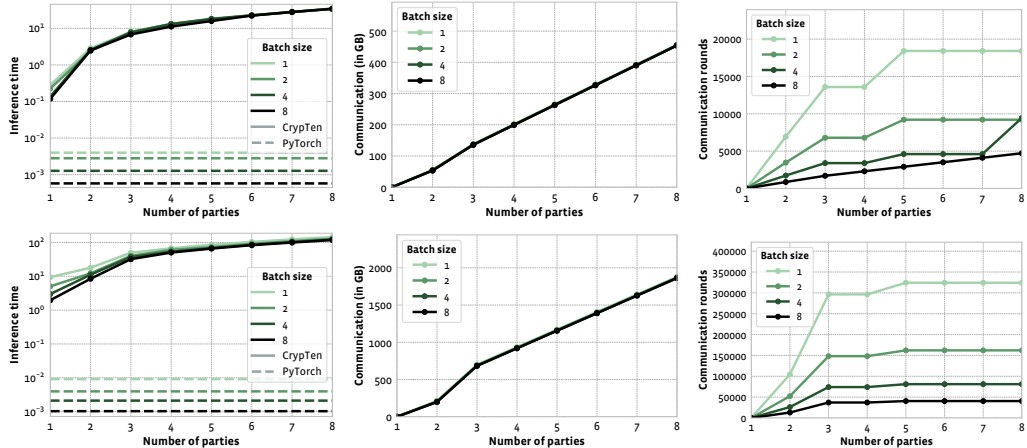

Figure 9: Benchmarks for inference with image-classification models on GPUs in CRYPTEN and PyTorch. **Top:** Results for ResNet-18 model. **Bottom:** Results for ViT-B/16 vision transformer. **Left:** Average wall-clock time per sample (in seconds). **Middle:** Number of bytes communicated per sample, per party (in GB). **Right:** Number of communication rounds per sample.

conversion between arithmetic and binary secret sharing and back (see the appendix). The number of communication rounds increases when the number of parties grows beyond $4$: CRYPTEN uses a tree reduction for the summation in the comparator protocol, which implies that the number of communication rounds grows whenever the number of parties increases from $2^k$ to $2^{k+1}$.

Figure 7 also presents results comparing Wav2Letter inference time between CPUs and GPUs. The results in the figure show that CRYPTEN is 1-2 orders of magnitude faster on GPUs than on CPUs. In real-world settings, this speedup can make the difference between a secure MPC use case being practical or not. Figure 8 shows how much wall-clock time is spent on communication and computation, respectively, when performing inference with Wav2Letter (using batch size 32). The results suggest that, whereas multi-party evaluation is compute-bound on CPU, it is communication-bound on GPU. On GPUs, 63% of the time is spent on communication in eight-party computation.

## 6.3  Image Classification

We performed image-classification experiments on the ImageNet dataset using residual networks (ResNets; [35]) and vision transformers (ViT; [26]).[6] We experimented with a ResNet-18 with 18 convolutional layers and with a ViT-B/16 model that has 12 multi-head self-attention layers with 12 heads each, operating on image patches of $16 \times 16$ pixels. Following common practice [35], we preprocess images by rescaling them to size $256 \times 256$ and taking a center crop of size $224 \times 224$.

Figure 9 presents the results of our image-classification benchmarks, which show that two parties can securely evaluate a ResNet-18 model in $2.49$ seconds and a ViT-B/16 model in $8.47$ seconds. A notable difference compared to the prior results is that the number of bytes communicated per sample is no longer reduced by increasing the batch size. The reason for this is that the vast majority of communication involves tensors that have the same size as intermediate activation functions: activation tensors are much larger than weight tensors in image-classification models. The amount of communication required to evaluate the ViT-B/16 model is particularly high due to the repeated evaluation of the softmax function in the attention layer of Transformers [65]. We also observe that in ResNet-18, the number of communication rounds grows faster than expected for larger batch sizes. The reason for this is that the carry-lookahead adder [21] used in the conversion from $[x]$ to $\langle x \rangle$ is very memory-intensive. When CRYPTEN runs out of GPU memory, it replaces the adder by an implementation that requires $O(|\mathcal{P}|)$ communication rounds (compared to $(\log_2 |\mathcal{P}|)$ for the carry-lookahead adder) but that requires less memory.

---

[6]We adopted the ResNet implementation from `torchvision` and the ViT implementation from `https://github.com/rwightman/pytorch-image-models`. ViT's normalized mean squared error is larger than for other models because our Gaussian error function approximation converges slowly; see Section **??**.

# 7 Conclusion and Future Work

In this paper, we have introduced and benchmarked CRYPTEN. We hope that CRYPTEN's flexible, machine-learning first API design and performance can help foster adoption of secure MPC in machine learning. We see the following directions for future research and development of CRYPTEN.

**Numerical issues** are substantially more common in CRYPTEN implementations of machine-learning algorithms than in their PyTorch counterparts. In particular, the fixed-point representation with $L$ bits of precision ($L = 16$ by default) is more prone to numerical overflow or underflow than floating-point representations. Moreover, arithmetic secret shares are prone to *wrap-around* errors in which the sum of the shares $[x]_p$ exceeds the size of the ring, $Q = 2^{64}$. Wrap-around errors can be difficult to debug because they may only arise in the multi-party setting, in which no individual party can detect them. We plan to implement tools in CRYPTEN that assist users in debugging such numerical issues.

**End-to-end privacy** requires seamless integration between data-processing frameworks, such as secure SQL implementations [5], and data-modeling frameworks like CRYPTEN. In "plaintext" software, such frameworks are developed independently and combined via "glue code" or platforms that facilitate the construction of processing and modeling pipelines. Real-world use cases of machine learning via secure MPC require the development of a platform that makes the integration of private data processing and modeling seamless, both from an implementation and a security point-of-view.

**Differential privacy** mechanisms may be required in real-world applications of CRYPTEN in order to provide rigorous guarantees on the information leakage that inevitably occurs when the results of a private computation are publicly revealed [28]. CRYPTEN implements sampling algorithms for the Bernoulli, Laplace, and Gaussian distributions (see appendix), which allows for the implementation of randomized response [68], the Laplace mechanism [29], and the Gaussian mechanism [6, 28] (although care must be taken when implementing these mechanisms [12, 46]). In future work, we aim to use these mechanisms, for example, to do a secure MPC implementation of DP-SGD [2].

**Threat models** may vary per use case. Specifically, some use cases may require malicious security or may not provide a TTP. Possible extensions may include support for malicious security via message authentication codes [22], as well as support for Beaver triple generation via additive homomorphic encryption [52], oblivious transfer [39], or more recent methods [10] to eliminate the need for a TTP.

**Model architecture design** for secure MPC is another important direction for future research. Following prior work in this research area, this study has focused on implementing *existing* machine-learning models in a secure MPC framework. However, these models were designed based on computational considerations in "plaintext" implementations of the models on modern GPU or TPU hardware. The results of our benchmarks suggest that this may be suboptimal because those considerations are very different in a secure MPC environment. For example, the evaluation of softmax functions over large numbers of values requires a lot of communication in secure MPC, which makes attention layers very slow. This implies that multilayer perceptron models [64] are likely much more efficient than vision transformers [26, 65] for image classification. We hope that CRYPTEN's machine-learning API and ease of use will spur studies that design model architectures specifically optimized for a secure MPC environment, for example, via neural architecture search [44, 47, 71].

# 8 Broader Impact

Although we believe that the adoption of secure MPC in machine learning can lead to the development of AI systems that are substantially more private and secure, we note that there are also potential downsides to such adoption. In particular, because the computations in secure MPC are performed on encrypted data, it can be harder to do quality control of AI systems implemented in CRYPTEN. For example, it is impossible to inspect the values of intermediate activations (or even model outputs) unless all parties agree to reveal those values. This may make it harder to explain why a model makes a certain decision [25] or to detect data-poisoning attacks [8]. Indeed, there exist fundamental trade-offs between privacy and utility [57] and those trade-offs apply to CRYPTEN users, too.

It is also worth noting that, although the protocols implemented in CRYPTEN come with rigorous cryptographic guarantees, practical implementations of these protocols may be broken by other means. For example, we have no reason to assume that CRYPTEN would not be susceptible to side-channel attacks [61]. Hence, good data stewardship remains essential even when using secure computation.

**Acknowledgments.** We thank Joe Spisak, Sijun Tan, Gregory Chanan, Igor Fedan, and the PyTorch team for their support. We thank Mark Tygert, Anderson Nascimento, Amrita Roy Chowdhury, and anonymous reviewers for helpful discussions and feedback on early versions of this paper.

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
