# Supplemental Material for CRYPTEN: Secure Multi-Party Computation Meets Machine Learning

**Brian Knott**    **Shobha Venkataraman**    **Awni Hannun**
**Shubho Sengupta**    **Mark Ibrahim**    **Laurens van der Maaten**
Facebook AI Research
{brianknott,shobha,awni,ssengupta,marksibrahim,lvdmaaten}@fb.com

## A  Detailed Description of Secure MPC Protocols

### A.1  Secret Sharing

CRYPTEN uses two different types of secret sharing: (1) arithmetic secret sharing [9] and (2) binary secret sharing [11]. Below, we describe the secret sharing methods for single values $x$ but they can trivially be extended to real-valued vectors $\mathbf{x}$.

#### A.1.1  Arithmetic Secret Sharing

CRYPTEN uses arithmetic secret sharing to perform most MPC computations. In arithmetic secret sharing, a scalar value $x \in \mathbb{Z}/Q\mathbb{Z}$ (where $\mathbb{Z}/Q\mathbb{Z}$ denotes a ring with $Q$ elements) is shared across $|\mathcal{P}|$ parties in such a way that the sum of the shares reconstructs the original value $x$. We denote the sharing of $x$ by $[x] = \{[x]_p\}_{p \in \mathcal{P}}$, where $[x]_p \in \mathbb{Z}/Q\mathbb{Z}$ indicates party $p$'s share of $x$. The representation has the property that $\sum_{p \in \mathcal{P}} [x]_p \mod Q = x$. We use a fixed-point encoding to obtain $x$ from a floating-point value $x_R$. To do so, we multiply $x_R$ with a large scaling factor $B$ and round to the nearest integer: $x = \lfloor Bx_R \rceil$, where $B = 2^L$ for some precision parameter, $L$. To decode a value, $x$, we compute $x_R \approx x/B$. Encoding real-valued numbers this way incurs a precision loss that is inversely proportional to $L$. Since we scale by a factor $B$ to encode numbers, we must scale down by a factor $B$ after every multiplication. We do this using the truncation protocol described below.

**Addition.** The addition of two secret-shared values, $[z] = [x] + [y]$, can be trivially implemented by having each party $p$ sum their shares of $[x]$ and $[y]$: each party $p \in \mathcal{P}$ computes $[z]_p \leftarrow [x]_p + [y]_p$.

**Multiplication.** To facilitate multiplication of two secret shared values, the parties use random Beaver triples [1], generated in an offline preprocessing phase. A Beaver triple of secret shared values $([a], [b], [c])$ satisfies the property $c = ab$. The parties use the Beaver triple to compute $[\epsilon] = [x] - [a]$ and $[\delta] = [y] - [b]$ and decrypt $\epsilon$ and $\delta$. This does not leak information if $a$ and $b$ were drawn uniformly at random from the ring $\mathbb{Z}/Q\mathbb{Z}$. The product $[x][y]$ can now be evaluated by computing $[c] + \epsilon[b] + [a]\delta + \epsilon\delta$, where $\epsilon$ and $\delta$ requires a round of communication among all parties. It is straightforward to confirm that the result of the private multiplication is correct:

$$[c] + \epsilon[b] + [a]\delta + \epsilon\delta = [a][b] + [x][b] - [a][b] + [y][a] - [b][a] + ([x] - [a])([y] - [b])$$
$$= [x][y].$$

Because this result holds for any linear function, $f(\cdot)$, of two variables for which the triple $(a, b, c)$ satisfies $c = f(a, b)$, we use the same procedure to perform matrix multiplication and convolution.

**Square.** To compute the square $[x^2]$, the parties use a Beaver pair $([a], [b])$ such that $b = a^2$. The parties compute $[\epsilon] = [x] - [a]$, decrypt $\epsilon$, and obtain the result via $[x^2] = [b] + 2\epsilon[a] + \epsilon^2$.

**Truncation.** A simple method to divide an arithmetically shared value, $[x]$, by a public value, $\ell$, would divide the share of each party by $\ell$. However, such a method can produce incorrect results

**Algorithm 1:** Private computation of the wrap count for an arithmetically shared value.

> **Input:** Arithmetic secret shared value $[x]$,
> Secret shared random value $[r]$ and its wrap count $[\theta_x]$.
>
> Compute: $[z] \leftarrow [x] + [r]$
> **for** $p \in \mathcal{P}$ **do**
>     Party $p$ computes: $[\beta_{xr}]_p \leftarrow ([x]_p + [r]_p - [z]_p)/Q$.
> **end for**
> Construct: $[\beta_{xr}] = \{[\beta_{xr}]_p\}_{p \in \mathcal{P}}$
> Decrypt: $z \leftarrow \text{reveal}([z])$
> Compute during decryption: $\theta_z \leftarrow (\sum_p [z]_p - z)/Q$.
> Compute: $[\eta_{xr}] \leftarrow z < [r]$
> Compute: $[\theta_x] \leftarrow \theta_z + [\beta_{xr}] - [\theta_r] - [\eta_{xr}]$

when the sum of shares "wraps around" the ring size, $Q$. Defining $\theta_x$ to be the number of wraps such that $x = \sum_{p \in \mathcal{P}}[x]_p - \theta_x Q$, indeed, we observe that:

$$\frac{x}{\ell} = \sum_{p \in \mathcal{P}} \frac{[x]_p}{\ell} - \frac{\theta_x}{\ell}Q \neq \sum_{p \in \mathcal{P}} \frac{[x]_p}{\ell} - \theta_x Q.$$

Therefore, the simple division method fails when $\theta_x \neq 0$, which happens with probability $x/Q$ in the two-party case. Many MPC implementations specialize to the $|\mathcal{P}| = 2$-party case and assume this probability is negligible [19, 22, 27]. However, when $|\mathcal{P}| > 2$ the probability of failure grows rapidly and we must account for the number of wraps, $\theta_x$. We do so by privately computing a secret share of the number of wraps in $x$, $[\theta_x]$. To this end, we define three auxiliary variables:

- $\theta_x$ represents the number of wraps produced by the shares of a secret shared variable $[x]$, such that $x = \sum_p [x]_p - \theta_x Q$, where $Q$ is the ring size.

- $\beta_{xr}$ represents the differential wraps produced between each party's shares of two secret shared variables, $[x]$ and $[r]$, such that $[x]_i + [r]_i \mod Q = [x]_i + [r]_i - [\beta_{xr}]_i Q$.

- $\eta_{xr}$ represents the wraps produced by two plaintext variables, $x$ and $r$, such that $x + r \mod Q = x + r - \eta_{xr}Q$.

We use these variable in Algorithm 1 to compute $[\theta_x]$. This approach is inspired by Algorithm 4 of [27], but extends to an arbitrary number of parties. The correctness of this algorithm can be shown through the following reduction:

$$
\begin{aligned}
z &= x + r - \eta_{xr}Q \\
\sum_p [z]_p - \theta_z Q &= \left(\sum_p [x]_p - \theta_x Q\right) + \left(\sum_p [r]_p - \theta_r Q\right) - \eta_{xr}Q \\
\sum_p [z]_p - \theta_z Q &= \left(\sum_p [x]_p + [r]_p\right) - (\theta_x + \theta_r + \eta_{xr})Q \\
\sum_p [z]_p - \theta_z Q &= \left(\sum_p [z]_p - [\beta_{xr}]_p Q\right) - (\theta_x + \theta_r + \eta_{xr})Q \\
\sum_p [z]_p - \theta_z Q &= \left(\sum_p [z]_p\right) - (\beta_{xr} + \theta_x + \theta_r + \eta_{xr})Q \\
\theta_x &= \theta_z + \beta_{xr} - \theta_r - \eta_{xr}.
\end{aligned}
$$

We then use $[\theta_x]$ to correct the value of the division by $\ell$:

$$\frac{x}{\ell} = [y] - [\theta_x]\frac{Q}{\ell} \quad \text{where} \quad [y] = \left\{\frac{[x]_p}{\ell}\right\}_{p \in \mathcal{P}}.$$

In practice, it can be difficult to compute $[\eta_{xr}]$ in Algorithm 1. However, we note that $\eta_{xr}$ has a fixed probability of being non-zero, irrespective of the number of parties. Indeed, regardless of the number of parties, we have $P(\eta_{xr} \neq 0) = x/Q$. In practice, we can therefore skip the computation of $[\eta_{xr}]$ and simply set $\eta_{xr} = 0$. This implies that incorrect results can be produced by our algorithm with small probability. For example, when we encode a real value $\hat{x}$ using a fixed-point encoding $x = \lfloor B\hat{x} \rfloor$, truncation will produce an error with probability $P(\eta_{xr} \neq 0) = \lfloor B\hat{x} \rfloor/Q$. This probability can be reduced by increasing $Q$ or reducing the precision parameter, $B$.

*Security proof.* One can show the security of Algorithm 1 by noting that the only information gained by an adversary is the revealed shares of $[z]$, which are indistinguishable from white uniform random noise because shares of $[r]$ are chosen to be uniformly random.

### A.1.2 Binary Secret Sharing

Binary secret sharing is a special case of arithmetic secret sharing that operates within the binary field $\mathbb{Z}/2\mathbb{Z}$. In binary secret sharing, a sharing $\langle x \rangle$ of a value $x$ is generated as a set of arithmetic secret shares of the bits of $x$ within the binary field. Each party $p \in \mathcal{P}$ holds a share $\langle x \rangle_p$ that satisfies $x = \bigoplus_{p \in \mathcal{P}} \langle x \rangle_p$. Because addition and multiplication modulo 2 are equivalent to binary XOR and AND operations, we can use bitwise operations on integer types to vectorize these operations.

Note that XOR and AND operations form a basis for the set of Turing-complete operations (via circuits). However each sequential AND gate requires a round of communication, which makes all but very simple circuits very inefficient to evaluate via binary secret sharing. In CRYPTEN, we only use binary secret sharing to implement comparators.

**Bitwise XOR.** Similar to addition in arithmetic secret sharing, a binary XOR of two binary secret-shared values, $\langle z \rangle = \langle x \rangle + \langle y \rangle$ can be trivially implemented by having each party XOR their shares of $\langle x \rangle$ and $\langle y \rangle$. That is, each party $p \in \mathcal{P}$ computes $\langle z \rangle_p \leftarrow \langle x \rangle_p \oplus \langle y \rangle_p$.

**Bitwise AND.** Since the bitwise AND operation is equivalent multiplication mod 2, we can utilize the same method we use to multiply arithmetic secret shared values. To facilitate bitwise AND of two binary secret-shared values, the parties use random triples generated in an offline preprocessing phase. The generated triple $(\langle a \rangle, \langle b \rangle, \langle c \rangle)$ satisfies the property $c = a \otimes b$. The parties then compute $\langle \epsilon \rangle = \langle x \rangle \oplus \langle a \rangle$ and $\langle \delta \rangle = \langle y \rangle \oplus \langle b \rangle$ and decrypt $\epsilon$ and $\delta$. This does not leak information since $a$ and $b$ contain bits drawn uniformly at random. $\langle x \rangle \otimes \langle y \rangle$ can now be evaluated by computing $\langle c \rangle \oplus (\epsilon \otimes \langle b \rangle) \oplus (\langle a \rangle \otimes \delta) \oplus (\epsilon \otimes \delta)$. Correctness follows from the same logic as multiplication in arithmetic secret sharing. We note that revealing $\epsilon$ and $\delta$ requires a round of communication among all parties in this protocol.

**Logical shifts.** Because each bit of a binary secret-shared value is an independent secret-shared bit, logical shifts can be performed trivially. To shift the bits of a binary secret-shared value $\langle x \rangle$ by a constant $k$, each party can compute the shift locally on its share, $\langle y \rangle_p = \langle x \rangle_p >> k$.

### A.1.3 Converting Between Secret-Sharing Types

Many machine-learning models require both functions that are easier to compute on arithmetic secret shares (*e.g.*, matrix multiplication) and functions that are easier to implement via circuits on binary secret shares (*e.g.*, argmax). Therefore, CRYPTEN uses both types of secret sharing and converts between the two types as needed using the techniques proposed in [10].

**From $[x]$ to $\langle x \rangle$:** To convert from an arithmetic share $[x]$ to a binary share $\langle x \rangle$, each party first secretly shares its arithmetic share with the other parties and then performs addition of the resulting shares. The parties construct binary secret shared values $\langle y_p \rangle$ where each $y_p$ represents one of the arithmetic secret shares $y_p = [x]_p$. This process is repeated for each party $p \in \mathcal{P}$ to create binary secret shares of all $|\mathcal{P}|$ arithmetic shares $[x]_p$. Subsequently, the parties compute $\langle x \rangle = \sum_{p \in \mathcal{P}} \langle y_p \rangle$. To compute the sum, a carry-lookahead adder circuit can be evaluated in $\log_2(|\mathcal{P}|) \log_2(L)$ rounds [5, 8]. In practice, the carry-lookahead adder circuit is quite memory-intensive. When CRYPTEN runs out of GPU memory, we adopt an alternative adder circuit that requires substantially less memory but performs $|\mathcal{P}| \log_2(L)$ communication rounds to perform the summation.

**From $\langle x \rangle$ to $[x]$:** To convert from a binary share $\langle x \rangle$ to an arithmetic share $[x]$, the parties compute $[x] = \sum_{b=1}^{B} 2^b \left[ \langle x \rangle^{(b)} \right]$, where $\langle x \rangle^{(b)}$ denotes the $b$-th bit of the binary share $\langle x \rangle$ and $B$ is the total number of bits in the shared secret. To create the arithmetic share of a bit, the parties use $b$ pairs of secret-shared bits $([r], \langle r \rangle)$ generated offline. Herein, $[r]$ and $\langle r \rangle$ represent arithmetic and binary secret-shares of the same bit value $r$. Parties then use Algorithm 2 to generate $\left[ \langle x \rangle^{(b)} \right]$ from $\langle x \rangle^{(b)}$. This process can be performed for each bit in parallel, reducing the number of communication rounds required for the conversion process to one.

---

**Algorithm 2:** Private single bit conversion from binary to arithmetic sharing.

---

**Input:** Binary secret shared bit $\langle b \rangle$; random bit in both arithmetic and binary sharing $[r], \langle r \rangle$.

Compute: $\langle z \rangle \leftarrow \langle b \rangle \oplus \langle r \rangle$.
Decrypt: $z \leftarrow \text{reveal}(\langle z \rangle)$.
Compute: $[b] \leftarrow [r] + z - 2[r]z$.

---

*Security proof.* One can show the security of Algorithm 2 by noting that the only information gained by an adversary is the revealed shares of $\langle z \rangle$, which are indistinguishable from white Bernoulli random noise because shares of $\langle r \rangle$ are chosen to be uniformly random.

### A.1.4 Logic-based Operations

Many applications require implementations of logic-based operators to make branching decisions and compute piece-wise functions.

**Comparisons.** To compare two secret-shared values $[x]$ and $[y]$, we can produce $[x < y]$ by computing their difference $[z] = [x] - [y]$ and comparing the result to zero: $[z < 0]$. We compute $[z < 0]$ by first converting $[z]$ to a binary secret-share $\langle z \rangle$, computing its sign bit using a right shift $\langle b \rangle = \langle z \rangle >> (L - 1)$, and converting the resulting bit to an arithmetic sharing $[b]$. Because we are using an integer encoding, the most significant bit of $z$ represents its sign. It is possible to compare $[x < y]$ directly using a less-than circuit, but this requires converting an extra value to binary secret sharing and incurring another $\log_2 L$ rounds of communication to compute the less-than circuit.

We can use the ability to compute $[x < y]$ to compute all other comparators on $[x]$ and $[y]$:

$$[x > y] = [y < x]$$
$$[x \geq y] = 1 - [x < y]$$
$$[x \leq y] = 1 - [y < x]$$
$$[x = y] = [x \leq y] - [x < y]$$
$$[x \neq y] = 1 - [x = y].$$

We optimize evaluation of the is-equal operator by computing $[x \leq y]$ and $[x < y]$ in parallel.

**Multiplexing.** Multiplexing is a very valuable tool for computing conditional and piece-wise functions. To multiplex between two values $[x]$ and $[y]$ based on a condition $c$, we must first evaluate $c$ to a a a binary value $[c] \in \{[0], [1]\}$. We can then compute $[c \ ? \ x : y] = [c][x] + (1 - [c])[y]$. This allows us to evaluate if-statements using CRYPTEN, where $[x]$ is the result when the if-statement is executed, and $[y]$ is the result otherwise. However, unlike if-statements, both results must be evaluated, meaning we cannot use tree-based or dynamic programming techniques to optimize algorithm runtimes.

**Sign, absolute value, and ReLU.** Several important functions can be computed using the multiplexing technique. We can compute $\text{sign}([x]) = 2[x > 0] - 1$. We can then use this to compute $|[x]| = [x]\,\text{sign}([x])$. Similarly we can compute the ReLU function by noting $\text{ReLU}([x]) = [x][x > 0]$.

**Argmax and maximum.** CRYPTEN supports two methods for computing maximums $[\max x]$. Both methods first compute a one-hot argmax mask that contains a one at the index containing a maximal element $[y] = \text{argmax}([x])$. A maximum can then be obtained by taking the sum $[\max x] = \sum_i [y_i][x_i]$ where the sum is taken along the dimension over which the maximum is being computed. By default, the argmax is computed using a tree-reduction algorithm, though configurations are available to use pairwise comparisons depending on network bandwidth / latency.

The *tree-reduction* algorithm computes the argmax by partitioning the input into two halves, then comparing the elements of each half. This reduces the size of the input by half in each round, requiring $O(\log_2 N)$ rounds to complete the argmax. This method requires order $O(\log_2 N)$ communication rounds, $O(N^2)$ communication bits, and $O(N)$ computation complexity.

The *pairwise* method generates a matrix $[A]$ whose rows are constructed by the pairwise differences of every pair of elements $\forall i \neq j : [A_{ij}] = [x_i - x_j]$. We then evaluate all comparisons simultaneously by computing $[A \geq 0]$. All maximal elements will correspond to columns whose elements are all greater than 0, so we can compute the argmax mask $[m]$ by taking the sum over all columns of

$[A]$. However, if more than one maximal element exists, this will result in a mask $[m]$ that is not one-hot. To make this one-hot we take a cumulative sum $[c]$ of $[m]$ and return $[c < 2][m]$ to return the index of the first maximal element. This method requires $O(1)$ communication rounds, $O(N^2)$ communication bits, and $O(N^2)$ computation complexity. In theory, because of constant-round communication, this method should be more efficient than the tree-reduction algorithm when the network latency is high.

**Argmin and minimum.** To compute minimums and argmins, we compute our argmax mask with a negated input: $[\mathrm{argmin}\, x] = [\mathrm{argmax}(-x)]$.

## A.2 Mathematical Approximations

Many functions are very expensive to compute exactly using only addition, multiplication, truncation, and comparisons. CRYPTEN uses numerical approximations to compute these functions, optimizing for accuracy, domain size, and efficiency when computed on secret shares. Each of these approximations has a specific domain over which the approximation converges well. One can modify the domain of convergence for certain functions using function-specific identities. For example, $\forall a \in \mathbb{R}$:

$$\ln(x) = \ln(ax) - \ln(a)$$
$$x^{-1} = a(ax)^{-1}$$
$$e^x = e^{x-a}e^a.$$

CRYPTEN also offers configurable parameters for protocol-specific optimizations, for example, custom initializations that improve convergence for iterative methods in a pre-specified input domain.

### A.2.1 Exponential, Sine, and Cosine

There are many well-known polynomial approximations for the exponential function, for example, the Taylor series, $e^x = \sum_{n=0}^{\infty} \frac{1}{n!} x^n$. However, because exponentials grow much faster than polynomials, the degree of the polynomial we would need to approximate the exponential function increases exponentially as the domain increases. Therefore, we instead use the limit approximation, which allows us to do repeated squaring very efficiently:

$$e^x = \lim_{n \to \infty} \left(1 + \frac{x}{2^n}\right)^{2^n}.$$

CRYPTEN can also use the repeated-squaring method to compute complex exponentials efficiently, which enables the computation of the sine and cosine functions:

$$\cos x = \Re(e^{ix})$$
$$\sin x = \Im(e^{ix}).$$

### A.2.2 Reciprocal

CRYPTEN uses Newton-Raphson iterations to compute the reciprocal function. This method uses an initial guess, $y_0$, for the reciprocal and repeats the following update:

$$y_{n+1} = y_n(2 - xy_n).$$

This will converge to $\lim_{n \to \infty} y_n = \frac{1}{x}$ quadratically as long as the initial guess $y_0$ meets the Newton-Raphson convergence criterion, which is $0 < y_0 < \frac{2}{x}$ for the above. By default, CRYPTEN uses:

$$y_0(x) = 3e^{0.5-x} + 0.003,$$

to initialize the approximation, which provides convergence on a large domain. This function was found by inspection and can be replaced by a user-defined value using CRYPTEN's configuration API. Because this method only converges for positive values of $x$, we compute the reciprocal using the identity $\frac{1}{x} = \frac{\mathrm{sgn}\, x}{|x|}$. (Note that square matrix inverses and Moore-Penrose inverses can be found using similar techniques given input matrices with singular values that meet the convergence criterion.)

### A.2.3 Square Root and Normalization

CRYPTEN uses Newton-Raphson iterations to compute square roots. However, the Newton-Raphson update formula for square roots, $y_{n+1} = \frac{1}{2}(y_n + \frac{x}{y_n})$ is quite inefficient to compute on secret shares. Instead, we use the much more efficient Newton-Raphson update formula for inverse square root:

$$y_{n+1} = \frac{1}{2} y_n (3 - xy_n^2).$$

We then multiply by the input $x$ to obtain the square root: $\sqrt{x} = (x^{-0.5})x$. We can also use the inverse square root function to efficiently normalize values via: $\frac{x}{\|x\|} = x \left( \sum_i x_i^2 \right)^{-1/2}$.

### A.2.4 Logarithm and Exponents

To compute logarithms, CRYPTEN uses higher-order iterative methods to achieve better convergence. The following update formula can be found using high-order modified Householder methods on $\ln(x)$ [26] or by manipulating the Taylor series expansion of $\ln(1 - x)$:

$$h_n = 1 - xe^{-y_n}$$

$$y_{n+1} = y_n - \sum_{k=1}^{\infty} \frac{1}{k} h_n^k.$$

Note that at each step $\ln x = y_n + \ln(1 - h_n)$, but we can only approximate $\ln(1 - h_n)$ using a truncated Taylor Series approximation. For this method, the order of the Householder method (*i.e.*, the polynomial degree in the second equation) will determine the speed of convergence. Since the convergence rate per iteration increases proportionally to the degree of the polynomial, whereas an exponential must be computed for each iteration, it is more computationally efficient to use high-degree polynomials instead of doing many iterations. By default, CRYPTEN uses a polynomial of degree 8, the initialization $y_0 = \frac{x}{120} - 20e^{-2x-1} + 3$, and 3 iterations. This provides effective convergence on the domain $[10^{-4}, 10^2]$.

Using the logarithm and exponential functions, we can also compute arbitrary public or private exponents on positive inputs $x$ using the equation $x^y = e^{y \ln(x)}$.

### A.2.5 Sigmoid and Hyperbolic Tangent

We have explored several methods for computing logistic functions in MPC, including direct computation, rational approximations, and Chebyshev polynomial approximations [12]. We have found that direct computation is the most efficient when it is combined with some specific optimizations. Specifically, CRYPTEN uses the exponential and reciprocal functions to compute:

$$\sigma(x) = \frac{1}{1 + e^{-x}}.$$

We optimize this computation by noting that the range of the sigmoid function is $[0, 1]$, and the range for the positive half of its domain is $[0.5, 1]$. Therefore, when we compute the reciprocal using the method described in Section A.2.2, we compute $\sigma(|x|)$ using an initialized value of $0.75$ for the Newton-Raphson iterations to improve convergence. We extend the result to the full domain by noting $\sigma(-x) = 1 - \sigma(x)$. We compute the hyperbolic tangent function via $\tanh(x) = 2\sigma(2x) - 1$.

### A.2.6 Gaussian Error Function

We use a Maclaurin series to approximate the Gaussian error function $\text{erf}(x) = \frac{2}{\sqrt{\pi}} \int_0^x e^{-x^2} dx$. The resulting approximation is given by: $\text{erf}(x) \approx \frac{2}{\sqrt{\pi}} \sum_{k=0}^{K} \frac{(-1)^k x^{2k+1}}{k!(2k+1)}$, where $K$ is the number of terms in the approximation (we set $K = 8$ by default). Although the approximation works reasonably well in practice, we note that it is known to have poor convergence when $x > 1$ (see OEIS A007680).

## A.3 Random Sampling

Several applications of privacy-preserving computations require secret-shared generation of random numbers such that no party can gain any information about the value of realizations. We use the following methods for generating secret shares of random samples from several popular distributions.

### A.3.1 Uniform Sampling

Due to quantization introduced by our encoding with scale $2^L$, we can only produce discrete uniform random variables with $2^L$ possible values. To do so, we produce samples $[u] \sim Uniform(0,1)$ by generating $L$ bits as Rademacher variates. To generate these bits, each party randomly generates its own binary secret-share with the same distribution locally. The XOR sum of independently distributed Rademacher variates, $u = \oplus_{p \in \mathcal{P}} \langle u \rangle_p$, is itself a Rademacher variate and is uncorrelated with any of the input bits.

*Security proof.* One can show the security of this sampler by noting that no adversary could gain any information about the sampled bit from its own binary share of the bit, because the XOR sum of independently distributed Rademacher variates is uncorrelated with any of the input bits. The bits are then converted to an arithmetic share $[u]$ using Algorithm 2, which is itself secure.

### A.3.2 Bernoulli Sampling

To compute a Bernoulli random variable with arbitrary mean $[b] \sim Bern(p)$, we first generate a uniform random variable $[u] \sim Uniform(0,1)$ and compute $[b] = [u > p]$. Note that due to quantization in $[u]$, the true probability parameter of the Bernoulli random sample is quantized to the nearest multiple of $2^{-L}$, as would have happened if $p$ was encoded using the fixed-point encoder.

### A.3.3 Gaussian Sampling

Gaussian random samples $[x] \sim \mathcal{N}(\mu, \sigma^2)$ can be computed using the Box-Muller transform. Given a pair of independent uniformly distributed random variables $([u_1], [u_2])$, two independent Gaussian random variables $([x_1], [x_2])$ from $\mathcal{N}(0,1)$ can be generated by computing:

$$[x_1] = \sqrt{-2\ln[u_1]}\cos(2\pi[u_2])$$
$$[x_2] = \sqrt{-2\ln[u_1]}\sin(2\pi[u_2]).$$

Since the range of the uniform inputs is $[0,1]$, we optimize our numerical approximations for better performance on this domain. To obtain samples $[y] \sim \mathcal{N}(\mu, \sigma^2)$, we compute $[y] = \sigma[x] + \mu$.

### A.3.4 Exponential and Laplace sampling

Exponential random variables $[x] \sim Exp(\lambda)$ can be computed using the inverse CDF method. Given a uniform random sample $[u] \sim U[0,1]$, an exponential random variable is generated via:

$$[x] = -\lambda^{-1}\ln([u]).$$

Again, we optimize the logarithm for the domain [0, 1].

A Laplace distributed random sample $[y] \sim Lap(\mu, k)$ can be generated from an exponential random sample, $[x] \sim Exp(k^{-1})$, and a Rademacher variate, $[b]$, by evaluating $[y] = (2[b] - 1)[x]$.

### A.3.5 Weighted Random Sampling

To produce a weighted random sample of inputs $[x_i]$ with weights given by $[w_i]$, we first generate a uniform random sample in $([0, \sum_i [w_i])$ by drawing a uniform sample, $[u]$, and evaluating $[r] = [u] [\sum_i w_i]$. Care should be taken to avoid precision issues caused by generating $[u]$ in fixed-point with finite precision. We then compute the cumulative sum values $[c_i]$ of the weights $[w_i]$, and compare those values to our random value $[m_i] = [c_i > r]$. This produces a mask vector whose entries are all zero below some index $j$ and all one above index $j$. To convert this mask vector into a one-hot vector, we append a zero in front of the $[m_i]$-values and compute $[o_i] = [m_i] - [m_{i+1}]$. Finally, we obtain the selected sample from the inputs $[x_i]$ by multiplying the samples with the one-hot vector and summing: $[y] = \sum_i [x_i][o_i]$.

## B  Comparison with Secure MPC Frameworks for Machine Learning

Table 1 presents a comparison of CRYPTEN with other secure MPC frameworks for machine learning. For each framework, the table shows whether the framework supports maliciously secure threat

models, can generate Beaver triples (if needed) without requiring a trusted third party, supports GPU computations, supports model training, supports general purpose function evaluation, and implements automatic differentiation (autograd). We define a secure MPC framework for machine learning to be general-purpose if it supports at least the following functions: linear functions, convolutions, rectified linear units (ReLU), max-pooling, and batch normalization.[1]

| Framework | Malicious security | Triple generation | Supports GPUs | Supports training | General purpose[†] | Supports autograd |
|---|---|---|---|---|---|---|
| *Two parties* | | | | | | |
| Chameleon [22] | ✗ | ✗ | ✗ | ✗ | ✗ | ✗ |
| Delphi [16] | ✗ | ✔ | ✗ | ✗ | ✗ | ✗ |
| EzPC [6] | ✗ | ✔ | ✗ | ✗ | ✗ | ✗ |
| Gazelle [13] | ✗ | ✔ | ✗ | ✗ | ✗ | ✗ |
| MiniONN [15] | ✗ | ✔ | ✗ | ✗ | ✗ | ✗ |
| PySyft [24] | ✗ | ✔ | ✔ | ✗ | ✗ | ✗ |
| SecureML [18] | ✗ | ✔ | ✗ | ✔ | ✗ | ✗ |
| XONN [23] | ✔ | N/A | ✗ | ✗ | ✗ | ✗ |
| *Three parties* | | | | | | |
| ABY3 [17] | ✗ | N/A | ✗ | ✔ | ✗ | ✗ |
| Astra [7] | ✗ | ✔ | ✗ | ✔ | ✗ | ✗ |
| Blaze [20] | ✗ | ✔ | ✗ | ✔ | ✗ | ✗ |
| CrypTFlow [14] | ✗ | N/A | ✗ | ✗ | ✔ | ✗ |
| CryptGPU[‡] [25] | ✗ | ✗ | ✔ | ✔ | ✔ | ✔ |
| Falcon [28] | ✔ | N/A | ✗ | ✔ | ✔ | ✗ |
| SecureNN [27] | ✗ | N/A | ✗ | ✔ | ✗ | ✗ |
| *Four parties* | | | | | | |
| FLASH [3] | ✔ | N/A | ✗ | ✔ | ✗ | ✗ |
| Trident [21] | ✔ | N/A | ✗ | ✔ | ✗ | ✗ |
| *Arbitrary number of parties* | | | | | | |
| CRYPTEN (ours) | ✗ | ✗[§] | ✔ | ✔ | ✔ | ✔ |

Table 1: Overview of secure MPC frameworks for machine learning and their properties. [†]We define a framework to be general-purpose if it supports all of the following layers: linear, convolution, rectified linear unit (ReLU), max-pooling, and batch normalization. [‡]We note that CryptGPU was developed *on top of* CRYPTEN, hence, it inherits many features from CRYPTEN. [§]Future versions of CRYPTEN will support Beaver triple generation without requiring a trusted third party.

## C  Overview of Functions Implemented in CRYPTEN

Table 3 gives an overview of all functions currently implemented in CRYPTEN, together with a short description of the approach used to implement the function. Random samplers are not listed in the table. For full details on the CRYPTEN secure-computation protocol, we refer to Appendix A.

## D  Security of CRYPTEN Functions

CRYPTEN provides MPC implementations of a large number of functions. However, these functions are all composed from a small set of primitives, which are listed in Table 2. CRYPTEN provides the security guarantee in Lemma 1. The proof for this security guarantee follows trivially from the following observations and results from prior work:

 i. Operations in arithmetic secret sharing are performed in the ring $\mathbb{Z}_{2^L}$. Multiplications in this ring are proven to be secure in [2].
 ii. Operations in binary secret sharing are performed using the GMW protocol [11]. AND operations in this protocol are proven to be secure in [10].
 iii. Conversion from arithmetic to binary (A2B) secret shares is performed using the protocol that is proven to be secure in [8].

---

[1]CRYPTEN supports a variety of functions beyond these five functions, but we focus on these five in our comparison as they are the main building blocks of many deep network architectures.

| MPC Primitive | Round Complexity | Security Proof |
|---|---|---|
| *Arithmetic secret sharing* | | |
| Addition | 0 | Non-interactive |
| Multiplication | 1 | [2, Theorem 1] |
| Truncation | $1^{\dagger}$ | Appendix A.1.1 |
| *Binary secret sharing* | | |
| XOR | 0 | Non-interactive |
| AND | 1 | [10, §III.B] |
| Bit-shift | 0 | Non-interactive |
| *Conversions* | | |
| A2B | $\log_2(|\mathcal{P}|)\log_2(L)$ | [8, §3] |
| B2A | 1 | Appendix A.1.2 |
| *Sampling* | | |
| Bernoulli(.5) | 1 | Appendix A.3.1 |

Table 2: Overview of the MPC primitives used in CRYPTEN, with their round complexity and references to the relevant security proof. Round complexity is defined as the number of sequential round-trips of communication required between parties to implement a given function, using an $L$-bit ring and $|\mathcal{P}|$ parties. $^{\dagger}$The number of rounds needed for truncation in the two-party setting is zero.

   iv. Tensor indexing operations like concatenation, selection, reshaping, *etc.* are non-interactive, which implies an adversary cannot gain any information.
   v. Security proofs for custom MPC protocols are provided in Appendix A (see Table 2 for details).
   vi. All other operations are compositions of secure functions (see Appendix A for details). This implies they are secure because security is closed under composition [4].

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

| Function | Function name(s) | Description |
|---|---|---|
| Absolute | abs | Multiply value by its sign. |
| Addition | add, + | Each party adds their shares. |
| Argument of maximum | argmax | Perform pairwise comparisons or tree reduction. |
| Argument of minimum | argmin | Perform pairwise comparisons or tree reduction. |
| Average pooling | avg_pool2d | Each party computes the average pooling of its share. |
| Batch normalization | batchnorm | Batch normalize values using summation, division, and variance functions. |
| Binary AND | and, & | Compute using binary Beaver protocol. |
| Binary cross-entropy | binary_cross_entropy | Compute using logarithm, multiplication, and addition functions. |
| Binary XOR | xor, ^ | Each party XORs it shares. |
| Clone | clone | Each party clones their share. |
| Comparison | >=, <=, =, ge, le, eq | To compare to $0$, convert to binary secret share and inspect most significant bit. |
| Concatenation | cat | Each party concatenates their shares. |
| Convolution | conv1d, conv2d | If filter is public, each party convolves its share. If filter is private, compute using Beaver protocol. |
| Cosine | cos | Approximate using repeated-squaring method. |
| Cross-entropy | cross_entropy | Compute using softmax, logarithm, multiplication, and division functions. |
| Cumulative sum | cumsum | Each party computes cumulative sum of values in its share. |
| Division | div, / | If divisor is public, divide shares by value and correct for wrap-around errors. |
| Dot product | dot | Multiply all elements and sum results. |
| Dropout | dropout | Each party multiplies their share with dropout mask. Dropout mask is not encrypted. |
| Error function | erf | Approximate using Maclaurin series. |
| Exponent | exp | Approximate using limit approximation. |
| Flatten | flatten | Each party flattens their share. |
| Flip | flip | Each party flips their share. |
| Hard tangent | hardtanh | Compute using comparison, multiplication, and addition functions. |
| Logarithm | log | Approximate using higher-order modified Householder method. |
| Log-softmax | log_softmax | Compute using exponentiation, maximum, summation, and addition functions. |
| Matrix multiplication | matmul | If one matrix is public, each party matrix-multiplies its share. If both matrices are private, compute using Beaver protocol. |
| Maximum | max | Compute argmax as one-hot vector; compute dot product with input. |
| Max pooling | max_pool2d | Compute maximum value. |
| Mean | mean | Each party computes mean of its share. |
| Minimum | min | Compute argmin as one-hot vector; compute dot product with input. |
| Multiplication | mul, * | If multiplier is public, each party multiplies its share with the multiplier. If multiplier is private, use Beaver protocol. |
| Multiplexing | where | Multiply first value by binary mask; add second value multiplied by inverse mask. |
| Negation | neg | Each party negates their share. |
| Norm | norm | Compute using the square, sum, and square root functions. |
| Outer product | ger | Perform multiplication of each pair of elements. |
| Padding | pad | Each party pads their share. |
| Permute | permute | Each party permutes their share. Indexes are not encrypted. |
| Product | prod | Multiply all elements in the input. |
| Power | pow, pos_pow | For positive powers, multiply in log-domain and exponentiate. For negative powers, compute reciprocal and evaluate positive power. |
| Reciprocal | reciprocal | Approximate using Newton-Rhapson iterations. |
| ReLU | relu, relu6 | Compare values with $0$, and multiply values by the resulting mask. |
| Reshaping | reshape, view | Each party reshapes their share. |
| Rolling | roll | Each party rolls their share. |
| Scattering | scatter | Each party scatters one share into the other share. Indexes are not encrypted. |
| Selection | gather, index_select, narrow, take | Each party selects part of their share. Indexes are not encrypted. |
| Sigmoid | sigmoid | Compute using the exponential and reciprocal functions. |
| Sign | sign | Compare value with $0$, multiply by 2, and subtract 1. |
| Sine | sin | Approximate using repeated-squaring method. |
| Softmax | softmax | Compute using exponentiation, maximum, summation, and reciprocal functions. |
| Square | square | Compute using Beaver protocol. |
| Square root | sqrt | Approximate using Newton-Rhapson iterations. |
| Squeezing | squeeze | Each party removes dimensions with size 1 from their share. |
| Stacking | stack | Each party stacks their shares. |
| Subtraction | sub, - | Each party subtracts their shares. |
| Summation | sum | Each party sums all values in its share. |
| Tangent | tanh | Perform linear transformation of sigmoid value of output. |
| Trace | trace | Each party sums all diagonal elements of their share. |
| Transpose | t, transpose | Each party transposes their share. |
| Unsqueezing | unsqueeze | Each party adds dimensions with size 1 to their share. |
| Variance | var | Compute using square, addition, and subtraction functions. |

Table 3: Overview of all functions on tensors implemented in CRYPTEN.