# OpenReview forum: "CrypTen: Secure Multi-Party Computation Meets Machine Learning"
_NeurIPS.cc/2021/Conference — NeurIPS 2021 Poster_

### Official Review · Reviewer_PL5J · 2021-07-16

**Rating:** 7
**Confidence:** 5

**Summary:**

The authors present an implementation for secure computation of deep learning inference and training. It is based on PyTorch, which makes its use easy for programmers familiar with the underlying framework. The secure computation is semi-honest and uses a trusted third party, which is assumed not to collaborate with any of the other parties.


**Limitations And Societal Impact:**

No concerns

**Main Review:**

I think the effort towards making secure computation more accessible deserves attention, and I appreciate the work the authors have done in this direction.

The authors hint at MPC compilers being potentially faster. What order of magnitude would that be? The authors refer to numerous works, but don't mention any results. For example, CrypTFlow takes 25.9 seconds and 6.9 GB for ResNet-50. In particular, the communication figure seems to be much better than the about 50 GB for ResNet-18 shown in this work.

The paper says that using a ripple-carry adder takes O(log(L)) rounds for L being the number of bits, which is wrong. Ripple-carry addition refers to the textbook addition that O(L) rounds. An adder taking O(log(L)) rounds would be a carry-lookahead adder.

I'm missing an argument in Section 5.3 for how much time using the GPU saves. What is the improvement per clock cycle or second?

The authors say that the results for private training are very similar to inference. I would expect that, in the context MPC, training fares better due to implicit parallelization. Is that what the authors mean here?

I consider the frequent announcement of future features ("we plan to add/support...") out of place in an academic paper, and I would prefer a more neutral style, say "a possible extension would be...".

Minor:
l244: Please move the URL to a footnote or the appendix.
l317: should be Newton-Raphson


**Time Spent Reviewing:**

3

---

> ### Author Response · Authors · 2021-08-09
>
> Thank you for your insightful comments and positive feedback on our work!
>
> MPC Compilers: Compilers have the potential to generate more efficient code because they know the computation graph before they execute it, and can apply optimizations to it. However, from past experience with deep-learning frameworks, it can be difficult for compilers to deliver on that potential: for example, TensorFlow (graph compiler) is not much faster than PyTorch (no graph compiler) for most practical use cases. Note that the decision on whether or not to use a compiler does not influence the amount of communication: the communication characteristics depend purely on the protocol implemented. CrypTFlow requires fewer bytes of communication because it implements a protocol that is specifically optimized for three-party inference. So the additional communication in CrypTen is the price we incur for supporting an arbitrary number of parties (we hopefully make up for it by using highly optimized primitives to actually implement that communication).
>
>
> Ripple-carry adder: Great catch! CrypTen implements the carry set/propagate/kill algorithm described in Section 6.3 of [21]. The log_2(|P|) log_2(L) complexity reported in the paper is correct, but we incorrectly referred to the algorithm as Ripple addition. Indeed, the algorithm is a carry-lookahead adder. We will make sure to correct this in the camera-ready version of the paper.
>
> Speed-up due to using GPUs: The exact improvement in wall-clock time due to offloading computations to the GPU depends on a lot of factors, including the model architecture, batch size, interconnect between GPUs / nodes, etc. This makes it difficult to make any general statements in Section 5.3. However, Figure 7 presents a direct comparison between Wav2Letter inference time on CPU and GPU. The results demonstrate that the speed-up can be an order of magnitude or more. The analysis in Figure 8 shows that, when computations are off-loaded to the GPU, the overall computation tends to become more communication-bound than compute-bound (even though communication is very fast in our experimental setup).
>
> Training versus inference: In CrypTen’s MPC protocol, training does not allow for much more parallelization than inference. Specifically, private multiplications involve performing an all-reduce, which is a blocking operation that tends to reduce the positive effects of additional parallelization. As a result, training and inference perform very similarly (training is roughly 2x slower than inference because it involves forward-backward passes rather than just forward passes). In CrypTen, parallelization can be improved primarily by increasing the batch size as illustrated in, for example, Figure 5.
>
> Phrasing of future features: We agree, the current phrasing is a bit awkward and probably not well-suited for an academic paper. We like your suggested rephrasing and will make sure to use it in the camera-ready version of the paper.

---

### Official Review · Reviewer_oKEp · 2021-07-16

**Rating:** 8
**Confidence:** 4

**Summary:**

This work is to fill the gaps of software framework for MPC used in Machine Learning. It provides many MPC primitives and fits them into ML scenario.

**Limitations And Societal Impact:**

No potential negative societal impact has seen.

**Main Review:**

Pros:
1. A great contribution to allow ML researcher, who may be lack of solid cryptography knowledge, have a chance to work on MPC-based privacy of ML with relatively easy access to implementation.
2. This work does more than basic ML by providing tensor computation, automatic differentiation and modular neural network to facilitate works deep learning with PyTorch-friendly APIs for state-of-the-art CV and NLP works.
3. With ability to support CrypTen on GPU, it achieves great performance on multiple benchmarks, such as image recognition and speech recognition.

Cons:
1. Although CrypTen did great jobs on passive security allowing an arbitrary number of parties, active security is also crucial to protect ML privacy, and it may helps more on real-world applications, since malicious adversary deviate cryptographic protocols with any arbitrary action.

**Time Spent Reviewing:**

3

---

> ### Author Response · Authors · 2021-08-09
>
> Thank you for your insightful comments and positive feedback on our work!
>
> Indeed, the current implementation and experiments consider the semi-honest setting. (We think we have stated this clearly in the paper, but please let us know if it is unclear.) We focused on this setting because we believe it is a reasonable threat model for many real-world use cases. In our experience, if there is no trust between parties, the parties are generally not willing to collaborate even if the MPC implementation is maliciously secure. Having said that, we have initial working implementations of malicious security via message authentication codes (MACs). This should provide CrypTen users to choose the threat model that is most appropriate for them.

---

### Official Review · Reviewer_ye4J · 2021-07-19

**Rating:** 6
**Confidence:** 4

**Summary:**

This paper presents a general software framework with GPU support for secure machine learning over multiple (arbitrary number of) parties: CrypTen. It provides a friendly way for people who don't have any cryptography background to implement the secure machine learning tasks, which could tolerate all-but-one corruptions in semi-honest case. It also demonstrates its capability and efficiency in realizing the state-of-the-art models for text classification, speech recognition, and image classification.

**Ethics Review Area:**

["I don’t know"]

**Limitations And Societal Impact:**

Those are properly addressed in the paper.

**Main Review:**

This paper is clearly written and technically sound. The experiments well demonstrates the performance of this software framework.

Strength: the software framework is easy to use and requires no knowledge of cryptography. It's the first public framework to realize secure modern machine learning tasks among arbitrary number of parties. And it shows a competitive performance for those tasks.

Weakness:
 - As stated in the paper, it only provides secure against semi-honest adversaries. While other works presents malicious security, but for 2 to 4 parties.
 - In the current implementation, the security relies on a Trusted Third Party.
 - Most of the techniques were already presented here: https://eprint.iacr.org/2019/599.pdf   I think the authors should also cite this paper.

**Time Spent Reviewing:**

8

---

> ### Author Response · Authors · 2021-08-09
>
> Thank you for your insightful comments and positive feedback on our work!
>
> Indeed, the current implementation and experiments consider the semi-honest setting, and we rely on a trusted third party to generate Beaver triples. (We think we have stated this clearly in the paper, but please let us know if it is unclear.) We focused on this setting because we believe it is a reasonable threat model for many real-world use cases. In our experience, if there is no trust between parties, the parties are generally not willing to collaborate even if the MPC implementation is maliciously secure. Having said that, we have initial working implementations of malicious security via message authentication codes (MACs), Beaver-triple generation via oblivious transfer (OT), and private multiplication via replicated secret sharing (RSS) that we will include in CrypTen shortly. This should provide CrypTen users with support for the most common threat models, and allow them to choose the most appropriate model for them.
>
> Thanks for pointing out that we are missing a reference to Damgard et al. (2019). We had included a reference to their work in a draft version of the paper, but it must have been inadvertently removed while editing the paper. We will make sure that the reference is there in the camera-ready version of the paper.

---

### Official Review · Reviewer_aZn5 · 2021-07-19

**Rating:** 8
**Confidence:** 4

**Summary:**

The authors proposed crypten, a new deep learning framework for secure multi-party computation (MPC) based on the PyTorch API. One of the important obstacle in adoption of MPC for deep learning is that using the existing MPC libraries requires extensive knowledge about MPC. A big advantage of crypten is that it follows the same familiar API of PyTorch and hides the MPC complexity behind the API. The provided code snippets demonstrate that using crypten is certainly easy for non-experts. They also support GPU acceleration for arithmetic computation.


**Limitations And Societal Impact:**

The authors discussed the limitation and shortcoming of the paper and their plan on how to extend the work. They also mentioned the possible broader impact of their work on the society and the community.

**Main Review:**

Originality:
- The authors use existing well-known MPC techniques and optimization to create the framework.
- Following the exact API of a well-known deep learning framework, like PyTorch, for a MPC library is novel.
- The authors could've done a better job in reviewing and comparing the existing MPC frameworks for deep learning, for example, a table comparing the features of different frameworks.


Quality:
- The code snippets and the experiments on different application sufficiently demonstrated crypten's ease-of-use.
- The shortcomings of the framework are also presented and explained.


Clarity:
- The paper is well written and organized and it is easy to follow.


Significance:
- The researchers and practitioners will find the work interesting and can probably build on top of the framework.
- The paper is the first to present a framework which follows a well-known deep learning API.


**Time Spent Reviewing:**

3

---

> ### Author Response · Authors · 2021-08-09
>
> Thank you for your insightful comments and positive feedback on our work!
>
> We like your suggestion of including a table comparing various characteristics of existing MPC frameworks for (deep) machine learning. We tried to give an overview of related frameworks in Section 2, but the current description is admittedly terse. Per your suggestion, we will include a more detailed description and corresponding overview table in the camera-ready version of the paper.

---

### Decision · Program_Chairs · 2021-09-27

**Decision:**

Accept (Poster)

**Comment:**

There is an agreement in the reviews that the paper’s secure multi-party computation (SMPC) framework for deep learning training and inference provides a valuable contribution. While most of the techniques are already known and the protection is only against honest-but-curious (as opposed to malicious) adversaries, the paper seems to fill an important gap and the contributions are a step forward towards broader deployment of SMPC protocols for ML primitives, and could stimulate future work in the area. I therefore lean towards acceptance.